# Structural Evolution in Glassy Carbon Investigated Based on the Temperature Dependence of Young’s Modulus

**DOI:** 10.3390/ma16134794

**Published:** 2023-07-03

**Authors:** Yi Yang, Yanpei Dang, Haihui Ruan

**Affiliations:** Department of Mechanical Engineering, The Hong Kong Polytechnic University, Hung Hom, Hong Kong; 20099079r@connect.polyu.hk (Y.Y.); dangyanpei@nimte.ac.cn (Y.D.)

**Keywords:** glassy carbon, Young’s modulus, pyrolysis, phenolic resin, structural reformation

## Abstract

As a non-graphitized carbon material, possessing exceptional hardness and chemical inertness, glassy carbon (GC) is often synthesized through the pyrolysis method, which includes a compression procedure of powdered precursor materials, thus increasing the costs for production of glassy carbon at an industrial scale. Direct preparation of GC via pyrolysis of bulk precursors is a low-cost approach but encounters challenges arising from an insufficient knowledge of carbon structure formation. In order to solve this problem, a new analysis of the temperature-dependent variation in Young’s modulus of GC obtained by the pyrolysis of phenolic resin at 1000 °C, utilizing the impulse excitation technique (IET), was performed. Our findings demonstrate that there is a critical temperature range of 500–600 °C where pyrolysis leads to the most significant density change and GC is formed as a result. When GC samples are heated again, a significant structural reformation occurs in the same temperature range. It causes a decrease in stiffness, especially at heating rates >3 °C/min, and an interesting restorative effect–increase in stiffness when a GC sample is annealed at temperatures of 500–550 °C. These results bring important implications for the direct formation of large amounts of glassy carbon using bulk precursors.

## 1. Introduction

Carbon materials have found broad applications in numerous industries, serving as essential components in mold tools for high-temperature glass molding and electrodes for batteries and fuel cells. However, due to their inherent low strength and brittleness, graphite-based products are fragile when deployed in operational conditions requiring high stresses and high temperatures. This fragility can result in significant economic losses and other challenges, such as the potential contamination that can arise from the dispersion of graphite powders. Hence, a non-graphitizing bulk carbon, named glassy carbon (GC), has received a lot of attention [1,2,3,4] since it was invented in the 1960s [5] through pyrolyzing of some cross-linked polymers containing oxygen interconnects (e.g., furan resin) or hydroxy side groups (e.g., phenol-formaldehyde resin (PFR)). GC has disordered microstructures with curled and randomly interlaced graphene layers [6] and exhibits physical and chemical properties superior to graphite. For example, GC has much higher thermal stability, chemical inertness, and hardness than graphite and possesses an electrical conductivity comparable with graphite [7]. The physicochemical properties exhibited by GC render it a highly desirable material for substitution of graphite in various applications. GC has been used mainly as a mold, crucible, and electrode material in high-temperature and highly corrosive environments. For example, Ju et al. [8] recently reported the preparation of GC molds with nanostructured surfaces and the use of them in a nanoimprint process to replicate the nanostructures from the GC surface to a glass surface. Similarly, Takahashi et al. [9] recently reported the successful use of GC molds to press Pyrex glass and quartz glasses with high fidelity and precision. Because GC is electric conductive and highly inert in aggressive chemical environments, and GC electrodes can be used to detect nicotinamide–adenine dinucleotid (NADH) [10] and H_2_O_2_ [11], which exhibit high stability and sensitivity. In addition, GC can also be used as an adsorption agent and as a dental implant [12]. In ref. [13], the wide application of GC has been reviewed.

However, the employment of GC products is constrained by their exorbitant cost, primarily attributed to low manufacturing yield and high production expenses. Additionally, owing to its high hardness and brittleness, GC cannot be easily machined, necessitating production methods analogous to those employed in the manufacture of ceramics, as reported by Jenkins and Kawamura in 1976 [14]. The process typically involves crushing a cured resin, usually brittle in nature, into fine powders, which are subsequently compacted, sintered, and then pyrolyzed to eliminate non-carbon atoms. Alternatively, GC can be directly pyrolyzed from a block of cured resin, potentially leading to the expansion of pores generated during curing due to the significant volume shrinkage and gas release that occurs during pyrolysis [4]. The process of pyrolysis results in the formation of internal defects, such as cavities and microcracks, that could induce fractures, particularly in cases where the thickness of a given section exceeds a certain threshold (e.g., larger than 6 mm [14,15]). However, the underlying failure mechanisms associated with these internal defects remain elusive, rendering it exceedingly challenging to identify optimal heating parameters to mitigate cracking tendencies in GC products with large thicknesses. This inadequacy represents a significant bottleneck in the development of a reliable fabrication process for GC, crucial for enhancing yield and reducing production costs.

Since GC was discovered in 1963 by Redfern et al. [5], several follow-up studies were immediately carried out to determine the best fabrication conditions for the commercialization of such products. For example, Jenkins et al. [14] classified the pyrolysis process into three stages (i.e., the pre-carbonization, carbonization, and dehydrogenation stages) based on the results of a thermogravimetric analysis (TGA) and the main reaction products in the three temperature ranges. Yamada et al. [16] investigated how pyrolysis temperature affected the physical and mechanical properties of GC. Recently, with the development of various characterization tools, researchers have been able to investigate the microstructural and compositional evolution within GC using various non-destructive analytical techniques, such as X-ray Diffraction (XRD) [3,6,17], Raman spectroscopy [3,17,18], Fourier-transform infrared (FTIR) spectroscopy [3,19,20], and electron microcopies, such as scanning electron microscope (SEM) [3,17] and transmission electron microscopes (TEM) [21,22]. There is also an increasing interest in employing GC to develop miniature devices for applications especially in high-temperature and highly corrosive environments, because some commercial photoresists (e.g., SU-8) can be converted into GC through pyrolysis, in which the volume shrinkage is an advantage for miniaturizing devices. GC has been used in microelectromechanical systems (MEMS) [13] as neural [2,23], cell [24], gas [25], and electrochemical [26] detectors. In addition, three-dimensional (3D) GC nanolattices (also known as mechanical metamaterials) were fabricated [1] to demonstrate GC nanostructures with the highest specific load carrying capacity among all materials [27].

The aim of this work is to synthesize glassy carbon through direct pyrolysis of bulk precursors. However, the abovementioned investigations on GC still cannot provide a clear understanding on how defects are developed when the precursor is converted to GC. This issue is particularly important for fabricating large volumes of bulk GC products, such as precision GC molds for glass molding [23] and GC materials in electrochemical systems used for replacing graphite. In order to have more understandings on GC and the associated pyrolysis process, in this work, the structure change in GC by probing its temperature-dependent Young’s modulus using the impulse exciting technique (IET) will be studied. 

## 2. Experimental Methods

The occurrence of cracks and pores during pyrolysis often leads to the fracture of a GC sample. We looked into this issue by conducting Thermal Gravity Analysis (TGA) and Thermal Mechanical Analysis (TMA) to probe the density change. We also employed a 3D Laser Scanning Microscope to characterize the precursor, GC surfaces, and impulse excitation technique (IET) to examine how the elastic property of GC, reflecting the density of internal defects, changes with temperature. In order to gain a deeper understanding of the carbon structures correlated with the property variations, X-ray Diffraction (XRD) patterns and Raman Spectroscopy of GC samples were measured. 

### 2.1. GC Fabricating Process

The flowchart of GC fabrication is as shown in Figure 1. In this work phenol formaldehyde (PF) resin (obtained from Henan Borun Casting Materials Co., 93 Ltd., Gongyi, China), with a room-temperature viscosity of 16–19 Pa·s and a carbon mass content of 40–45%, was used as a bulk precursor for direct pyrolysis to produce glassy carbon. The resin can be cured in an incubator at 100 °C after being degassed in a low vacuum environment for 1 h. The curing time was varied from 6 to 22 h to examine its influence. The cured precursor was pyrolyzed in a tube furnace with an N_2_-purged atmosphere. In pyrolysis, a GC precursor is heated from room temperature to 1000 °C at a rate of 1 °C/min. In the heating process, we hold the temperatures at 300, 400, 500, 600, and 1000 °C for 60 min in order to drive away gaseous reaction products at different stages. After that, the pyrolyzed product is naturally cooled in the furnace. 

### 2.2. Characterizations

We employed various characterization tools to investigate the variation in GC structure in thermal cycles, including TGA, Thermomechanical analysis (TMA), IET, XRD, Raman spectroscopy, 3D laser scanning microscope, and an electrical conductivity analyzer, which are introduced hereunder. 

#### 2.2.1. Thermogravimetric Analysis (TGA) and Thermomechanical Analysis (TMA)

Thermal gravimetric analysis (TGA) was performed in a nitrogen environment using a Rigaku STA8122 thermogravimetric analyzer. The GC samples were heated at a rate of 5 °C/min, from room temperature to 800 °C. To study the pyrolysis process we adopted, cured PF samples were protected in argon gas at a flow rate of 100 mL/min and heated from room temperature to 1000 °C at a rate of 3 °C/min and were then held at 300, 400, 500, and 600 °C for one hour each.

We employed TMA to investigate the dimension change during pyrolysis. The experimental set-up before and after the TMA test is illustrated in Figure 1. We prepared a cured cylindrical PF sample with a diameter of 4.600 mm and a height of 2.844 mm for pyrolysis in the TMA, as shown in Figure 2a. The sample was protected in flowing (100 mL/min) argon gas and subjected to a temperature ramp-up from room temperature to 1000 °C at a rate of 3 °C/min followed by natural cooling. During heating, the sample was held at 300, 400, 500, and 600 °C for one hour each. During the TMA test, a quartz probe with a circular cross-section was continuously applied with a force of 0.05 N onto the sample. Upon completion of the pyrolysis process, the GC shown in Figure 2b was obtained.

#### 2.2.2. Impulse Excitation Technique (IET)

The GC samples we prepared for IET (IMCE RFDA HT1600, Belgium) tests have the nominal dimension of 40 × 8 × 2 mm^3^ with a tolerance of ± 0.1 mm. In the IET furnace, a GC sample is hung at the two stationary nodes of its primary vibration mode to measure Young’s modulus as a function of temperature change. The detailed set-up is shown in Figure 3. A pneumatically actuated tapper strikes the midspan of the sample. This impulse causes the flexural vibration of the GC sample with sound generation, which is transmitted through a ceramic tube in the furnace and recorded by a microphone outside the furnace. The primary vibration frequency *f* of the GC sample is then obtained and Young’s modulus of the sample can be determined by [28]:(1)$$E=0.9465mf^{2}\frac{l^{3}}{wh^{3}}$$
where *m*, *w*, *l*, and *h* are the mass, width, length, and thickness of the sample, respectively. As the GC samples in this work were obtained by pyrolyzing PFR at 1000 °C, we expect that its service temperature will be below the maximum pyrolysis temperature. For example, GC can be used to fabricate molds [29] for precision glass molding, which is generally performed at temperatures below 800 °C. Hence, we heat the GC samples to 800 °C and naturally cool them to investigate the variation of *E*. 

#### 2.2.3. X-ray Diffraction (XRD) and Raman Spectroscopy

An X-ray diffractometer (Rigaku SmartLab 9kW, Tokyo, Japan) with an X-ray tube type of rotating anode type was used to perform the XRD measurements at a potential of 45 kV and a current of 200 mA. During the experiments, each XRD process scanned the 2θ range from 10 to 80°, with 0.01° step-size and an integration time of 10 seconds per point, lasting approximately 10 min in total.

The Raman scattering spectra of the GC samples were investigated in the spectral range of 100–1800 cm^−1^ at room temperature. The nonpolarized Raman spectra were recorded in the backscattering geometry using a Renishaw micro-Raman system with a 532 nm excitation light. The model was a Renishaw micro-spectrometer RL532C50 with a 532 nm edge filter. Spectra were obtained with 0.5% of the laser beam power and an exposure time of 20 s. A Leica 50× LWD microscope objective lens equipped with a numerical aperture (NA) of 0.5 was used to focus the laser beam on the sample surface. This results in a laser beam diameter of about 2 µm. An excitation power of 1 mW was fixed in order to prevent any damage to the sample. A spatial resolution of approximately 1 µm could be achieved using the Raman spectrometer. This experiment had a spectral resolution of 0.3 cm^−1^.

## 3. Results and Discussion

### 3.1. TMA and TGA Results of Pyrolysis

Pyrolysis induces volume and mass changes associated with internal structural evolution. Hancock et al. [30] measured the room-temperature density of GC precursors after pyrolyzing PF resins to various temperatures to infer the structural evolution. Here, we made an attempt to determine the in situ changes in volume, mass, and density during pyrolysis by using TMA and TGA, and the results are shown Figure 4. It can be observed that the pyrolysis resulted in a total decrease of 600 μm (21% of the original height) in the sample height and a total mass reduction of 40%. The sample height increased due to thermal expansion and decreased due to reactions. The quick increase in sample height associated with mass reduction in the temperature ranges of 140–150 and 210–220 °C were noted. At temperatures below 230 °C, a series of dehydration and condensation reactions occurred in the precursor, as described in [31]. The gaseous products escaped from the precursor leading to the formation of micropores. Our TMA and TGA results suggested that the reactions at 140–150 °C and 210–220 °C were radical, resulting in an obvious mass reduction but volume increase. From 220 to 300 °C and then during the one-hour dwelling at 300 °C, the sample height decreased, which corresponds to the pre-carbonization stage. We then observed another increase in the sample height from 300 to 400 °C, which should have been caused by thermal expansion and this expansion was associated with approximately 10% mass loss. This indicates that the pre-carbonized substance became more porous than the cured one in this temperature range. After 400 °C, the sample heigh only decreased and the fastest decrease occurred between 400–500 and then at 500 °C. 

With a pressing force of 0.05 N, the compressive stress during pyrolysis is approximately 3 kPa. We hypothesize that the stress was small and could not induce viscous deformation, i.e., the dimensional changes are assumed to be isotropic and the change in sample height can be used to determine the volume change. At the four holding temperatures, the changes in volume, mass, and density are then shown in Figure 5. It is noted that the time-dependent mass and volume changes at the four holding temperatures can be overlapped, indicating that the volume and mass reductions followed the same trend. Note that the initial density of the cured PF precursor at room temperature was 1.3 g/cm^3^. During the holdings, the calculated densities were all smaller than the initial room-temperature value. The density of the sample remained largely unchanged at 300 and 400 °C during the holding phase. During the one-hour holding at 500 and 600 °C, the increases in density were 0.123 g/cm^3^ and 0.085 g/cm^3^, respectively, which is remarkable. It implies a radical structural evolution in the temperature range of 500–600 °C. In the later study with GC samples, it will be shown that such a structural evolution in the temperature range of 500–600 °C is still significant when a GC sample is heated.

### 3.2. Surface Morphology Evolution after Pyrolysis 

GC prototypes with various geometries were fabricated using the method described in Section 2.1. The reliability of this preparation method is demonstrated in Figure 6, wherein different GC products are shown. In particular, the prototype shown in Figure 6a was prepared for subsequent IET tests. To examine the surface morphology and roughness of the GC prototypes, we cured two sets of precursors for 6 and 22 h. A 3D laser scanning microscope was employed to probe the surface morphology of the precursors and the corresponding GC specimens. It was found that the precursors exhibited small spots on the surface as highlighted in the red circles in Figure 7a,c, which were caused by bubbles in which gas did not escape during the degassing process due to the surface tension of the resin before curing. During the pyrolysis process, a certain number of grooves were generated on the surfaces of both samples. Our observations indicated that the GC obtained from the precursor cured for 6 h had fewer surface grooves than that obtained from the precursor cured for 22 h. It is noted that additional tiny holes and pits were observed in the GC obtained from the precursor cured for 6 h. However, this is not the case for the GC sample obtained from the precursor cured for 22 h. 

The distribution of surface grooves in the GC pyrolyzed from the 22 h cured precursor was more uniform, and the depth of surface grooves was slightly smaller, leading to a smaller increase in roughness from the precursor. The longer curing time led to the more crosslinking structures inside the precursor, resulting in a denser material [32]. As a result, the gaseous molecules escaped more slowly, which, combined with the gaseous formation and shrinkage during pyrolysis, led to stresses inside the bulk samples. These internal stresses are probably the main reason for the more branched surface grooves in the GC samples pyrolyzed from the precursor with the longer curing time [33]. Roughness of the sample surfaces are listed in Table 1. In the table, R_a_ describes the average of absolute deviation of asperity heights from an average value and R_q_ the root mean square of asperity heights deviating from the average value [33]. After pyrolysis, the GC obtained from the 6 h cured precursor exhibited a roughness increase of 93.4% and 148.3% for Ra and Rq, respectively. The GC obtained from the 22 h cured precursor exhibited a roughness increase of 73.0% and 61.3% for Ra and Rq, respectively. Pyrolyzing the precursors with a longer curing time led to a relatively smaller change in surface roughness, as manifested by the more homogeneous distribution of surface grooves.

### 3.3. Comparison between IET and TGA 

The room-temperature Young’s modulus of our GC samples was measured to be 21.5 GPa, very close to the values (21.7 GPa [34] and 21.37 GPa [35]) determined by nanoindentation. However, Sakai et al. [36] employed a resonance method to determine the Young’s moduli of two types of GC samples pyrolyzed at 2000 and 2500 °C, which were 28.0 and 30.3 GPa, respectively. Garion et al. [37] employed a four-point bending experiment to measure the Young’s modulus of two kinds of commercial GC samples after heat treatments at 1000 and 2000 °C. Their measurements were 32.4 and 32.5 GPa, respectively. Liu et al. [17] reported a Young’s modulus value of 27.64 GPa for a commercial GC specimen using the IET measuring method. It is noted that GC samples pyrolyzed at higher temperatures (over 2000 °C) have generally higher Young’s Modulus than our results and commercial GC generally belongs to the former category. Such a discrepancy is mainly due to the difference in the atomic arrangements as discussed in [38]. Moreover, incomplete pyrolysis, which is particularly the case for a low-temperature (~1000 °C) process, leaves impurities within a GC sample. These impurities can adversely affect the stiffness of GC due to their influence on the average crystallite size and vacancy distributions [13,39], especially during thermal cycles. To address these issues, Figure 8a shows the results of the IET experiments conducted with a GC sample obtained from the precursors cured for 22 h. In this set of IET experiments, the heating rate was 5 °C/min and we conducted three thermal cycles with a GC specimen. It is noted that the Young’s modulus of the GC sample varied with temperature and decreased with the number of cycles.

It is known that a heat treatment for GC can reduce or close internal micro-pores, contributing to an increase in elastic modulus [40]. Meanwhile, heating leads to thermal expansion and the anharmonicity results in a decrease in elastic modulus [41]. Therefore, the Young’s modulus of the bulk GC varies nonmonotonically with temperature, as demonstrated in Figure 8a, owing to the competition between the two mechanisms. In the first thermal cycle, the heating process lead to two humps in the modulus curve, which occurred in the temperature range of 100–200 °C and 500–600 °C, respectively. However, in the cooling process, these two humps vanished. In the second and third heating, the low-temperature (100–200 °C) hump also vanished. However, the high-temperature hump (500–600 °C) became much more significant than that of the first cycle. Notably, in pyrolyzing the precursor, the fastest density increase also occurred between 500 and 600 °C [32,42]. Hence, we conceive that the remarkable increase in the Young’s modulus of the GC sample can be correlated with the chemical process in this temperature range, for example, the remaining impurities in the GC sample may be driven away and carbon substructures may be more connected. It is noted that the most significant decrease in the Young’s modulus followed in the temperature range of 600–650 °C, suggesting a change in the crystalline structure in the temperature range of dehydrogenation during pyrolysis. 

We conducted TGA for a GC sample by employing the same heating cycles as those demonstrated in Figure 8a. The TGA results are shown in Figure 8b. In the first thermal cycle, the mass reduction was 2.20%, indicating that the original GC sample still contained non-carbon impurities. The mass change in the second and third cycles became much smaller, i.e., 0.39% and 0.2%, respectively. In the first heating process, the mass continuously decreased, hence the structural change in the temperature range of 500–600 °C was masked by the mass reduction. In the second and third cycles, the mass reduction in this temperature range was insignificant. Instead, a slight increase could be discerned, which was probably caused by the adsorption of argon atoms from the environment. The main point is that the significant mass reduction occurred between 600 and 650 °C followed by a much gentler mass reduction until 800 °C, corresponding well to the IET results. In addition, the mass did not change further during the cooling process. Such a correspondence indicates that the increase in elastic modulus in the range of 500–600 °C is caused by the rearrangement of carbon structures. 

### 3.4. XRD and Raman Results

The results of IET and TGA indicate the different tendencies in structural and compositional changes at temperatures before and after 600 °C, respectively. To understand more of such changes, we conducted heat treatment to the GC samples with a maximum temperature of 550 and 650 °C by still using the IET furnace with a heating rate of 5 °C/min. The samples before and after heat treatment were subjected to XRD and Raman analysis, leading to the results shown in Figure 9. 

The XRD patterns exhibited two broad peaks at angles of 24° and 42°, corresponding to the reflection from the crystallographic plane of (0 0 2) and a turbostratic band (1 0) [6,43]. The structural characteristic peaks of diamond are due to (100), (220), (311), and (400) reflections, while those of graphite are mainly from (0 0 2) reflections. Unlike these two carbon allotropes, the characteristic peaks of glassy carbon are ascribed to the (1 0) and (002) reflections [44]. The microstructure of glassy carbon is formed by interlaced graphene layers. $L_{c}$ is the average stack thickness of graphene layers in the c-direction and can be determined from the FWHM of (0 0 2) peak. $L_{a}$ is the mean layer diameter in the a-direction and can be determined from the FWHM of (1 0) peak [14]. It is worth noting that the XRD results for both samples indicate a slight rightward shift of the two peaks after the heat treatment. Based on the Scherrer equation, L=Kλ/βcos⁡θ, one can derive $L_{a}$ and $L_{c}$, where *K* is the shape factor, λ the wavelength of X-ray, and β the full width at half maximum (FWHM) of a peak [17]. It is found that the $L_{c}$ and $L_{a}$ reduce by 7.5% and 22.1%, respectively, after the heat treatment at 550 °C. For the heat treatment at 650 °C, the reductions in $L_{c}$ and $L_{a\;}$ are 4.3% and 6.8%, respectively. The position of the (0 0 2) reflection corresponds to the average spacing between carbon layers, $h_{002}$, which can be determined based on the Bragg’s equation: $h_{002}=\lambda /2sin\theta_{002}$ [43]. Consequently, the rightward shift of the XRD peaks indicates a decrease in the spacing of atomic planes. 

The results of Raman spectroscopy, as shown in Figure 9c,d, demonstrate that the intensities of the D and G bands at approximately 1340 cm^−1^ and 1580 cm^−1^, respectively, changed after the heat treatments. The D band is caused by the symmetrical stretching vibration (radial breathing mode) of the sp^2^ carbon atoms in aromatic hydrocarbon rings. Such a breathing mode becomes more active with more defects. Therefore, the intensity of the D band can be used to determine the extent (or number density) of the defected hydrocarbon rings, which has been explained in [45]. The G band is caused by the in-plane bond-stretching vibration of C=C bonds [46]. It is usually conceived that the ratio of the intensity of the D and G bands, denoted by *I*_D_/*I*_G_, indicates the significance of defects inside a graphitic or turbostratic carbon structure [47]. Figure 9c shows the results of the Raman analysis of a GC sample before and after heat treatment at 550 °C. We fitted the Raman analysis results with a Lorentz function and calculated *I*_D_/*I*_G_ from the peak height ratios. It is noted that *I*_D_/*I*_G_ is changed from 1.047 to 0.934 after heating. The results of heat treatment at 650 °C are shown in Figure 9d; similarly, *I*_D_/*I*_G_ is changed from 1.040 to 0.974. These reductions in *I*_D_/*I*_G_ suggest that the heat treatments lead to less in-plane defects in carbon sheets. 

Combining the XRD and Raman results, we conceive that the heat treatment results in less in-plane defects and inter-plane spacing. The cause of the decrease in inter-plane spacing could be that the reaction of impurities in the GC samples produces gaseous products. The leaves of non-carbon atoms bring about purer carbon layers with smaller inter-plane spacings. However, the transport of gaseous products also caused the reduction in crystallite dimensions (i.e., $L_{c}$ and $L_{a\;}$) and left more mesoscale defects, which is probably the main cause of the reduction in Young’s Modulus after the thermal cycles.

### 3.5. The Effect of Heating Rate

Thermal shocks are often the cause of failure when the heat flux or temperature gradient is too large. Hence, we changed the heating rate in the IET experiments to investigate if the degradation in stiffness, as shown in Figure 8a, is temperature or heat flux dependent. We prepared four GC samples, pyrolyzed from 22 h cured precursors, and heat them to 600 °C with heating rates of 3, 4, 6, and 10 °C/min. We set total heating time for the four samples to be the same and let the sample heated faster stay at 600 °C longer. The results, as shown in Figure 10, demonstrate that the Young’s modulus of the samples decreases more rapidly with increasing heating rate especially in the temperature range of 250–400 °C indicating that the larger heat flux causes the more significant structural damage even in a low-temperature regime corresponding to the pre-carbonization stage in pyrolysis. It should be noted that when the heating rate is 3 °C/min, the loss in the measured Young’s modulus becomes insignificant. These results suggested that the GC pyrolyzed at low temperatures (~1000 °C) should avoid the applications with fast heating or thermal shock. 

The XRD patterns of the four samples are shown in Figure 11. Again, the two peaks are rightward shifted, and the higher heating rate leads to the greater shift, suggesting the larger heat flux leads to the smaller spacing between carbon layers resulting from the more thorough reaction. Figure 10 and Figure 11 indicate that the Young’s modulus of the GC sample exhibits a gradual decline with the increase in the heating rate from 3 to 10 °C/min and that the XRD peaks shift rightward. These two phenomena are both attributed to an increase in the carbon layer spacing. The Raman results, as shown in Figure 12 and Table 2, indicate the same trend, where *I*_D_/*I*_G_ decreases more significantly when the heating rate is larger, suggesting less in-plane defects. However, the connectivity between the randomly arranged carbon layers becomes poorer after faster heating. This has been demonstrated by the IET results. In addition, the loss in electric conductivity, as shown in Table 3, further corroborates this argument. 

### 3.6. Thermal Enhancement

Figure 8a shows a hump in the variation of Young’s modulus with temperature, which maximizes at approximately 550 °C. It is conceived that such an inherent structural change may be utilized to stabilize the stiffness of GC especially after it is damaged by a thermal shock. To validate this concept, a new GC sample was prepared from 22 h-cured resin. It was first heat treated based on the temperature profile of the sample GC-6 demonstrated in Figure 10. This sample was then reheated to 550 °C at a rate of 5 °C /min for three cycles, and the corresponding results are demonstrated in Figure 13. In the first cycle, a significant reduction in Young’s modulus before 500 °C occurs again, while the modulus rises between 500 and 550 °C, implying a restorative effect on stiffness. In the second and third cycles, the Young’s modulus becomes very stable, with a slight increase during cooling caused by the restorative effect between 500 and 550 °C. 

## 4. Conclusions

Taken together, this work investigated the thermal degradation of GC prepared by low-temperature (1000 °C) pyrolysis that is often used in preparing various miniaturized GC devices or structures. The degradation is caused by heat flux which promotes internal reactions and the transport of gaseous products. The latter leads to a reduction in in-plane defects and smaller spacing between carbon layers, but smaller and more separate crystallinities. Such a structural change causes the loss in stiffness and electric conductivity. It is noted that higher heating rate causes more damage to GC samples; hence, GC is vulnerable to large temperature gradient or thermal shock. In this study, we employed the IET to investigate the temperature-dependent changes in the Young’s modulus of GC samples. It revealed a remarkable structural evolution in GC between 500–600 °C which leads to an increase in Young’s modulus. This suggests that an appropriate heat treatment to GC (e.g., annealing at 550 °C) may help in recovering the stiffness of the material.

## Figures and Tables

**Figure 1 materials-16-04794-f001:**
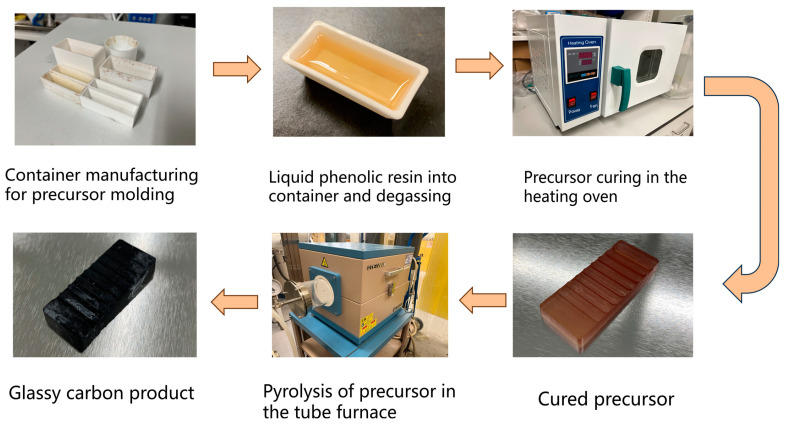
The flowchart of GC fabrication process.

**Figure 2 materials-16-04794-f002:**
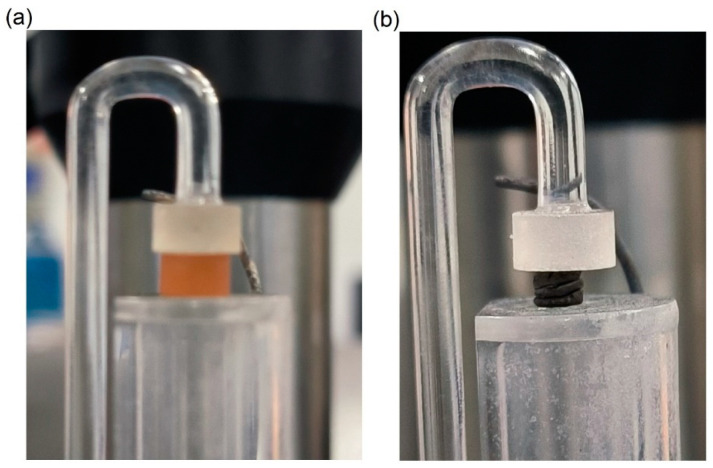
Experimental set-up of TMA tests: (**a**) before pyrolysis and (**b**) after pyrolysis.

**Figure 3 materials-16-04794-f003:**
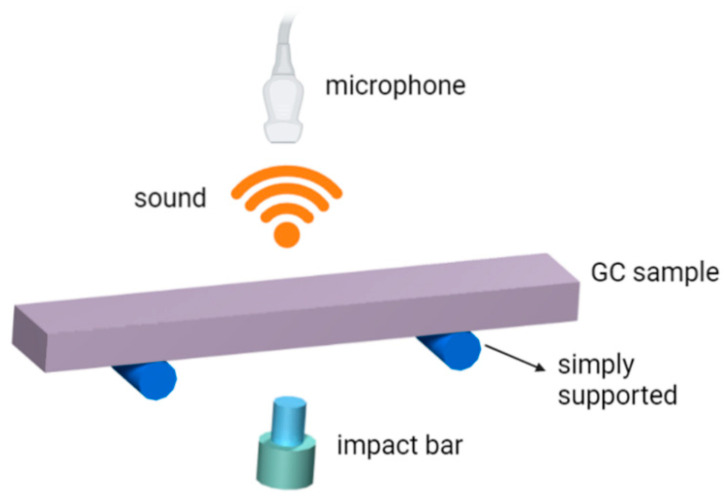
Schematic illustration of impulse excitation technique (IET).

**Figure 4 materials-16-04794-f004:**
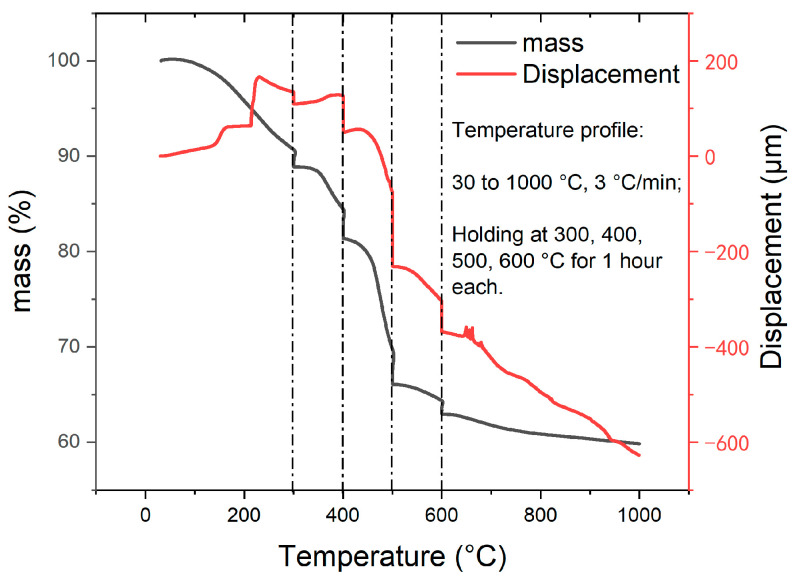
TMA and TGA results of pyrolysis to fabricate GC.

**Figure 5 materials-16-04794-f005:**
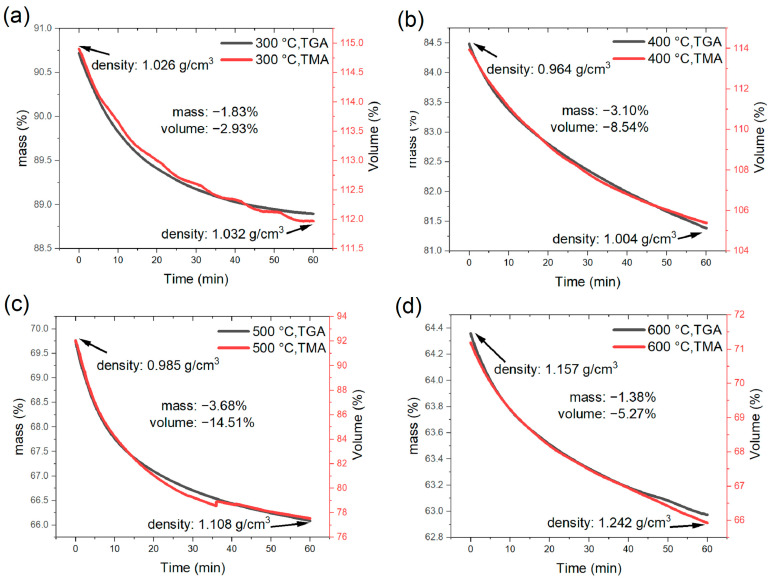
Volume, mass, and density changes at four holding temperatures: (**a**) 300 °C, (**b**) 400 °C, (**c**) 500 °C, and (**d**) 600 °C.

**Figure 6 materials-16-04794-f006:**
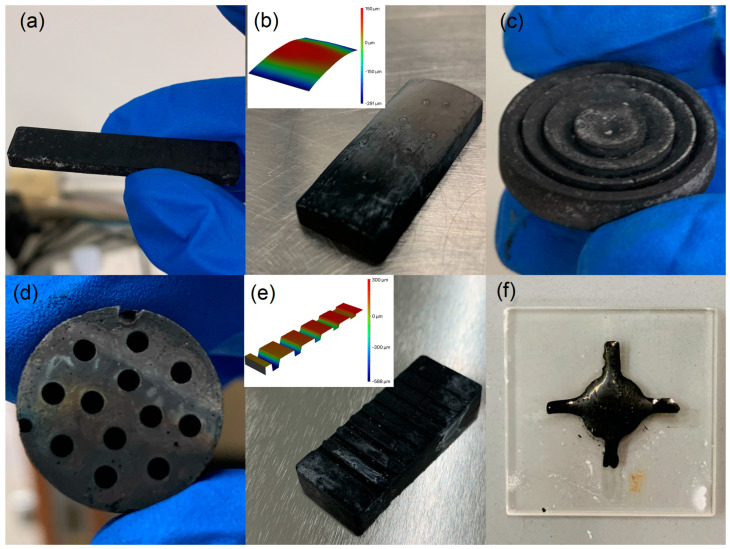
GC prototypes with various geometries: (**a**) a rectangular bar for IET testing, (**b**) curved surface, (**c**) Fresnel lens mold, (**d**) honeycomb, (**e**) surface gratings, and (**f**) a cross on a silica slide.

**Figure 7 materials-16-04794-f007:**
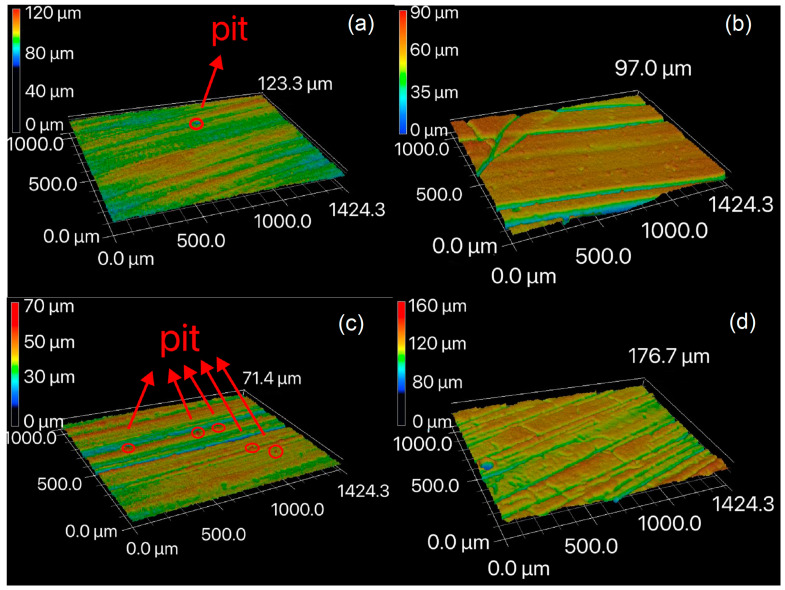
Surface topologies of (**a**) 6 h-cured precursor, (**b**) GC obtained from 6 h-cured precursor, (**c**) 22 h-cured precursor, and (**d**) GC obtained from 22 h-cured precursor.

**Figure 8 materials-16-04794-f008:**
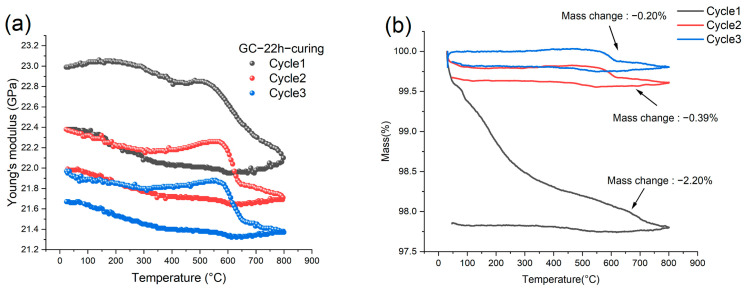
(**a**) The variation of Young’s modulus of a GC sample prepared by pyrolyzing 22-h-cured precursor in three thermal cycles with a heating rate of 5 °C/min and (**b**) Corresponding TGA results.

**Figure 9 materials-16-04794-f009:**
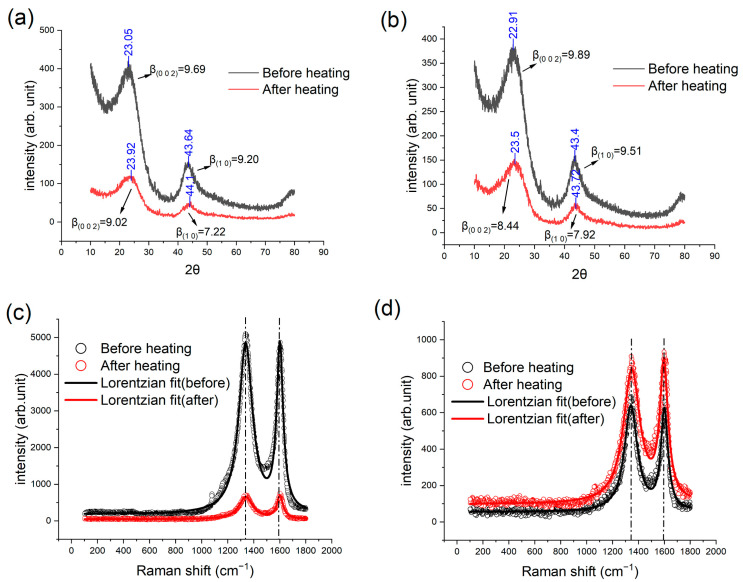
XRD (**a**,**b**) and Raman (**c**,**d**) results of GC sample after heat treatment at (**a**,**c**) 550 °C and (**b**,**d**) 650 °C.

**Figure 10 materials-16-04794-f010:**
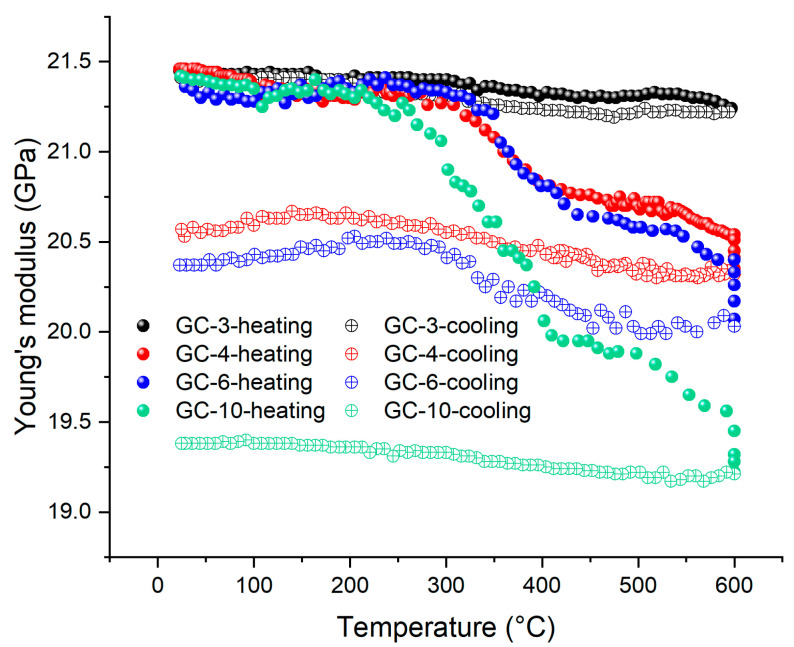
IET results of four GC samples subject to different heating rates (e.g., GC-3 means a heating rate of 3 °C/min).

**Figure 11 materials-16-04794-f011:**
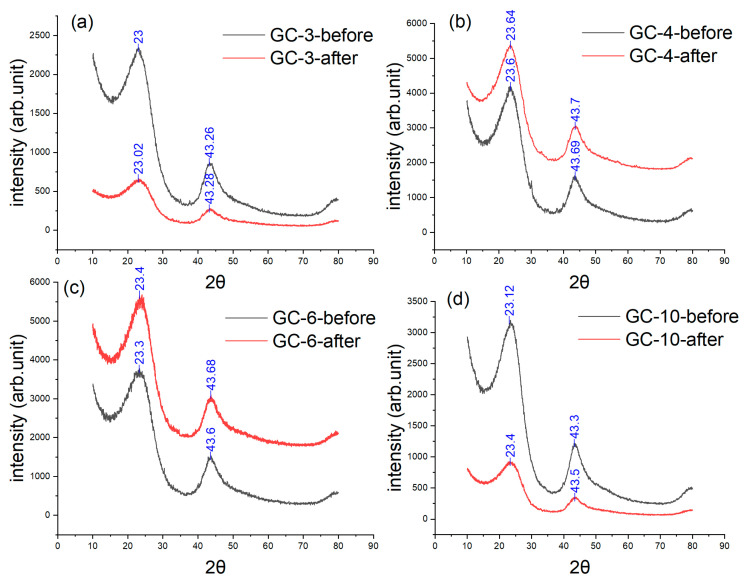
XRD results of four GC samples subject to different heating rates: (**a**) 3 °C/min, (**b**) 4 °C/min, (**c**) 6 °C/min, and (**d**) 10 °C/min.

**Figure 12 materials-16-04794-f012:**
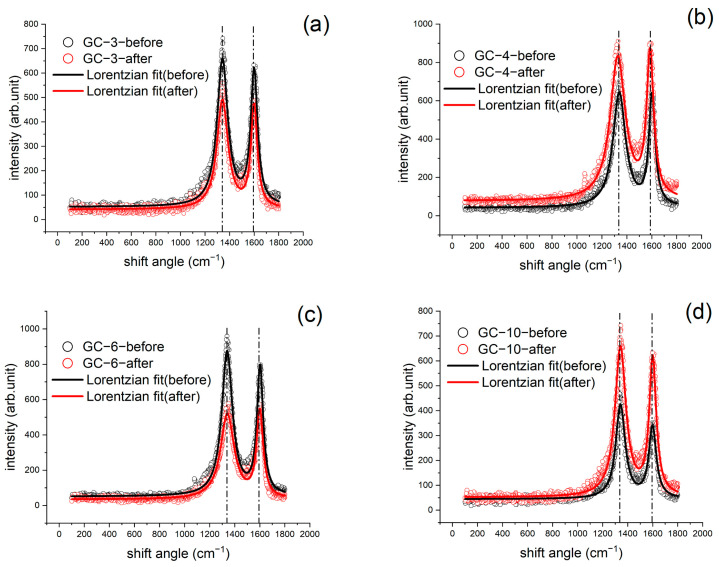
Raman results of four GC samples subject to different heating rates: (**a**) 3 °C/min, (**b**) 4 °C/min, (**c**) 6 °C/min, and (**d**) 10 °C/min.

**Figure 13 materials-16-04794-f013:**
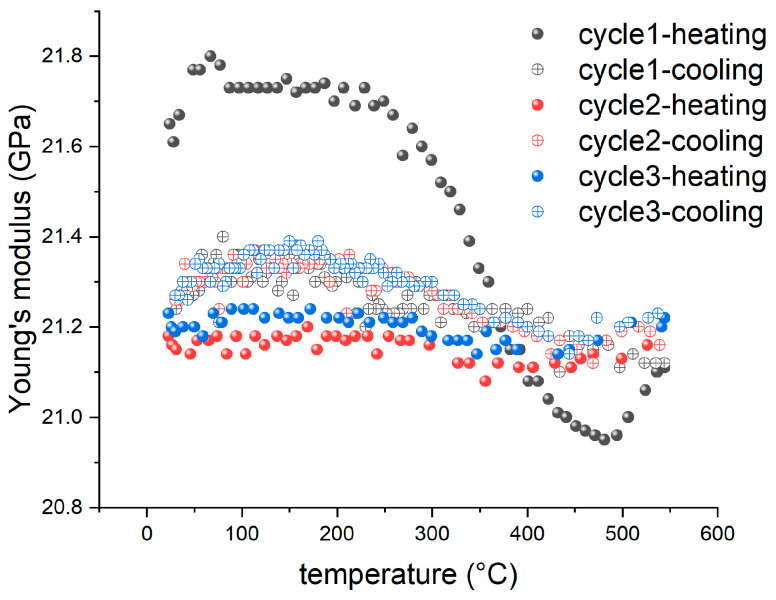
IET results of a GC sample annealed at 550 °C. The sample was preheated to 600 °C that causes thermal degradation.

**Table 1 materials-16-04794-t001:** Surface roughness R_a_ and R_q_ of precursors and GC samples.

Surface Roughness	R_a_ (μm)	R_q_ (μm)
6 h-cured precursor	4.1396	5.1740
GC obtained from 6 h-cured precursor	8.9007	12.8469
22 h-cured precursor	4.4794	5.9644
GC obtained from 22 h-cured precursor	7.7515	9.6199

**Table 2 materials-16-04794-t002:** Raman spectrum results (*I*_D_/*I*_G_) of GC samples subject to different heating rates.

Raman Results (*I*_D_/*I*_G_)	GC-3	GC-4	GC-6	GC-10
Before IET	1.1726	1.1099	1.180	1.329
After IET	1.1436	1.0120	1.010	1.107

**Table 3 materials-16-04794-t003:** Electrical conductivity of GC samples subject to different heating rates.

Electrical Conductivity(S/cm)	GC-3	GC-4	GC-6	GC-10
Before IET	99.133	105.51	108.502	110.313
After IET	53.345	40.324	35.336	N/A

## Data Availability

No new datasets were generated or analyzed during the current study.

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
