# Peer review of "Structural Evolution in Glassy Carbon Investigated Based on the Temperature Dependence of Young’s Modulus"

_materials, 2023, doi:10.3390/ma16134794_

Round 1
Reviewer 1 Report
Review of the Manuscript ID Materials-2421308
1. English text should be corrected. Abstract, Highlights, Introduction, Results and Conclusions should be rewritten.
Abstract
“As a non-graphitized carbon material possessing exceptional hardness and chemical inertness, glassy carbon (GC) is frequently synthesized through compressed pyrolysis of powdered precursor materials, which makes GC products highly pricy. Direct preparation of GC via pyrolysis of bulk precursors is a low-cost approach but encountered challenges arisen from the insufficient knowledge of carbon structure formation. To address this issue, here, we present an analysis of the temperature-dependent variation in Young's modulus of GC obtained by the pyrolysis of phenolic resin at 1000°C, utilizing the impulse excitation technique (IET). Our findings demonstrate a critical temperature range of 500 - 600 °C, in which the pyrolysis leads to the most significant density change in forming GC. When GC samples are reheated, a significant structural reformation occurred in the same temperature range. It causes decrease in stiffness especially when high heating rates are higher than 3 C/min, and an interesting restorative effect manifested by an increase in stiffness when a GC sample is annealed at a temperature between 500 – 550 °C. These results bear important implications for the direct formation of larger sizes of GC from bulk precursors.“
Should be replaced by
“As a non-graphitized carbon material, possessing exceptional hardness and chemical inertness, glassy carbon (GC) is often synthesized through pyrolysis method, which includes compression procedure of powdered precursor materials, thus increasing the costs for production of glassy-carbon in industrial scale. Direct preparation of GC via pyrolysis of bulk precursors is a low-cost approach but encountered challenges arisen from the insufficient knowledge of carbon structure formation. In order to solve this problem, new analysis of the temperature-dependent variation in Young's modulus of GC obtained by the pyrolysis of phenolic resin at 1000°C, utilizing the impulse excitation technique (IET), was performed. Our findings demonstrate that there is critical temperature range of 500 - 600 °C, where pyrolysis leads to the most significant density change and GC is formed as a result. When GC samples are heated again, a significant structural reformation occurred in the same temperature range. It causes decrease in stiffness, especially at heating rates > 3oC/min, and interesting restorative effect - increase in stiffness when a GC sample is annealed at temperatures 500 – 550 °C. These results bring important implications for the direct formation of large amount of glassy carbon using bulk precursors.:”
2.
Highlights are too long. Please, correct them according to journal requirements.
3.
Introduction, in the beginning
“However, owing to their inherent low strength and brittleness, graphite-based products are fragile when deployed in operational contexts requiring high stresses and high temperatures.”
Should be substituted with
“However, due to their inherent low strength and brittleness, graphite-based products are fragile when deployed in operational conditions requiring high stresses and high temperatures.”
4.
Introduction, at the end
“However, the abovementioned investigations on GC still cannot provide a clear understanding on how defects are developed when a GC-precursor is converted to GC, an issue particularly important for fabricating macroscopic GC products, such as precision GC molds for glass molding [23] and GC structures in electrochemical systems replacing graphite. In order to have more understandings on GC and the associated pyrolysis process, in this work, we study the structure change in GC by probing its temperature-dependent Young’s modulus using the impulse exciting technique (IET).”
Should be replaced by
“The aim of this work is synthesize glassy-carbon through direct pyrolysis of bulk precursors. However, the abovementioned investigations on GC still cannot provide a clear understanding on how defects are developed when the precursor is converted to GC. This issue is particularly important for fabricating large volumes of bulk GC products, such as precision GC molds for glass molding [23] and GC materials in electrochemical systems used for replacing graphite. In order to have more understandings on GC and the associated pyrolysis process, in this work, the structure change in GC by probing its temperature-dependent Young’s modulus using the impulse exciting technique (IET) will be studied.”
5.
|”Our main findings are that a fast heating process can deteriorate internal structures in GC resulting a decrease in stiffness and that there is a critical temperature range between 500 – 550 C in which a high stiff GC structure is formed.”
This sentence should be omitted and transferred to Conclusions section.
6. page 15, row 387
“Conclusion” should be corrected to “conclusions”.
7. Please correct the references according to journal requirements. Some references do not contain all the authors, others are without pages, volumes, etc.
8.
Experimental methods
“We aim to obtain GC through direct pyrolysis of bulk precursors. In this work, we 92
used phenol formaldehyde (PF) resin (obtained from Henan Borun Casting Materials Co., 93
Ltd), of which the room-temperature viscosity is 16-19 Pa.s with a carbon weight content 94
of 40-45%.”
Should be substituted with
“In this work phenol formaldehyde (PF) resin (obtained from Henan Borun Casting Materials Co., 93
Ltd), with room-temperature viscosity of16-19 Pa.s and carbon mass content of 40-45%, was used as bulk precursor for direct pyrolysis to produce glassy carbon.”

English text should be corrected. Abstract, Highlights, Introduction, Results and Conclusions should be rewritten.
Author Response
Please kindly have a check about the responds and revised manuscript,thanks.

Reviewer 2 Report
The manuscript titled “Investigation of the temperature-dependent Young’s modulus of glassy carbon” address the problem of mechanical properties of glass carbon obtained by phenolic resin pyrolysis. The manuscript is well written and the methods of investigations are adequately choosen. I think that the main finding described in the manuscript is glass carbon structural evolution between 500-600C which results in improvement in Young’s modulus. By this treatment the carbonaceous material may exhibit improved stiffness.
The small remark is for XRD analysis. The X-ray tube type should be given in experimental section. Also “a.u.” abbreviation is advised for astronomical units. Therefore the arb. unit is more appropriate (e.g. in Fig. 8). I think also that additionaly to 3D laser scanning microscope, the scanning electron microscopy would be also beneficial.
Taking into account these small remarks I think that this work will contribute to carbonaceous material knowledge and is worth of publication in MDPI Materials.
Author Response

(The authors gave the same response as above.)

Reviewer 3 Report
Dear Authors,
My comments are as follows:
1. The title must be improved, it looks a straightaway case study once you look into the title.
2. Introduction is well organized, add the relevant recent models no matter what the dependencies are. (Optional)
3. I must appreciate the experimentation part but it is not presented well. No flowchart, no categorization etc. You Just started to explain the process. (Revise thoroughly)
4. Does Fig 3 predicts the phenomenon? or can it be applied to other model if you are interested in mass/displacement stats? And why it is getting so closer? Fig 4 needs a better explanation since it carries useful information and looks quite improved.
5. Revisit Fig. 9/10 and try to summarize it.
Moderate changes.
Author Response

(The authors gave the same response as above.)

Reviewer 4 Report
Manuscript materials-2421308 “Investigation of the temperature-dependent Young’s modulus of glassy carbon” reports on the heat treatment of PF-derived glassy carbon at various temperatures and heat rates. Rigorous structural investigations are carried out, as well as in situ analyses of the Young’s modulus temperature-dependent variation via IET. The manuscript is mostly well-written, and the obtained results and conclusions are supported by the provided discussions. However, several major issues should be addressed. My concerns regarding the manuscript are mostly related to the discussion of Raman spectra provided by authors.
Comments:
1) Line 68-69, “within GC using various spectroscopies, such as X-Ray Diffraction (XRD)”: as XRD is not a spectroscopic study, consider revising this fragment.
2) Section 2.2.3: what model of the Renishaw spectrometer was used? What optical density filter was applied during the measurements of the Raman spectra and what acquisition time was set?
3) Line 169, by “The quick increases”, did you mean “The quick increase”?
4) Lines 169-171, “The quick increases in sample height associated with mass reduction in the temperature ranges of 140 -150 and 210 – 220 °C were noted, which was probably caused by fast cross-linking processes” – to me, it is unclear how cross-linking and mass reduction can lead to the increase of the height of carbonized polymer. If some functional groups of the polymer are removed, and resulting carbon backbones of various polymeric chains are being cross-linked forming some dense structure, why doesn’t the material shrink? Does the pore formation take place during the material removal, similarly to the dehydrohalogenation procedure?
5) In Fig. 4, title of the right Y axes in all subfigures should be “Volume”, not “Volumn”.
6) In Fig. 6, consider increasing the font of the numbers located near the scale bar. Currently, they are not visible.
7) Line 303-305: consider communicating on if the XRD lines are attributed to diamond or graphite. Their diffraction patterns are considerably different [10.1063/1.333385].
8) Line 304, by “(1 0)”, did you mean “(100)”?
9) Lines 303-310: to me, it is unclear how the Lc and La values were obtained (FWHM of which peak was used to estimate each of the values). Consider supplementing the discussion to communicate how one peak can reflect crystallite width and the other one represents crystalline height.
10) Line 315, “The results of Raman” → “The results of Raman spectroscopy”.
11) Lines 316-319, “The D-band arises from out-of-plane vibration which becomes more significant with more defects in crystalline structure, and the G-band represents the in-plane stretching of sp2 hybridization of carbon atoms.”: as I see it, D-line is attributed to the breathing mode of graphitic hexagons, while G-line is ascribed to the stretching vibration of C=C-bonds. Please supplement your discussion with the appropriate references. Consider the discussion related to Fig. 4 in [10.1103/PhysRevB.61.14095].
12) In lines 315-316, you mention that D- and G-lines shift after the annealing. However, you don’t discuss how the positions of the lines are changed and how such shift can be explained in terms of structural rearrangement. Additionally, although in glassy carbon the amount of the sp2-hybridized carbon may vary in 95-100% range [10.1007/s10853-017-1753-7], authors don’t provide any discussion on the sp2/sp3 ratio, which variation trend can be assessed from the position of G-line. Please supplement the discussion.
13) Fig. 8(c), black line, shows a pronounced shoulder at 1200 cm-1. Can it be related to the presence of polyenes (such as polyacetylene) in the structure? This feature of the spectra also raises the following questions: how were the spectra fitted to obtain ID/IG, what kind of the lines were used (Gaussian, Lorentzian etc.), why fitting lines and fitting results are not presented in the paper? Current presentation doesn’t allow to verify the data related to the ID/IG.
14) In Fig. 9, please indicate the parts of the curves related to heating and cooling. Currently the “directions” of the lines are unclear.
15) Table 3, GC-10 after IET: please discuss why the value couldn’t be obtained. Was it too low to be reliably measured?
Comments 3 and 10 are related to the minor issues related to wording. Overall, I think that the language of the maunscript is really good and entirely understandable.
Author Response

(The authors gave the same response as above.)

Round 2
Reviewer 3 Report
My suggestions have been responded well.
Check out some guidelines for abstract and conclusion.
Author Response
Please kindly check the response ,thanks.

Reviewer 4 Report
Authors have adequately answered the questions and resolved most of the issues related to the manuscript. Several additional comments are related to the new text and data.
1) Line 109, “Pa.s” typo should be fixed.
2) Line 195, by “leading to microporosity.”, did you mean “leading to the formation of micropores.”?
3) In Fig. 9(c,d) and Fig. 12, (Raman spectroscopy), please provide the legend denoting that solid curves are related to the fitting results.
4) Lines 359-360, phrase “ID/IG, is proportional to the significance of defects inside a carbon structure” is unclear. It also seems too specific, as, unlike NV-centers photoluminescence line in diamond, ID is not proportional to the number of the defects. Consider rephrasing.
5) Line 362, D-line positioning at 1300 cm-1 seems quite unconventional, as 1320–1360 cm−1 range is typical for D-line [10.3390/magnetochemistry8120171]. Could you comment on that?
Author Response
Please kindly check the response and the revised manuscript,thanks.
